# Current Situation of Bacterial Infections and Antimicrobial Resistance Profiles in Pet Rabbits in Spain

**DOI:** 10.3390/vetsci10050352

**Published:** 2023-05-14

**Authors:** Mercedes Fernández, Biel Garcias, Inma Duran, Rafael A. Molina-López, Laila Darwich

**Affiliations:** 1Departament de Sanitat i Anatomia Animal, Universitat Autònoma de Barcelona (UAB), 08193 Cerdanyola del Vallès, Spain; mariamercedes.fernandez2@autonoma.cat (M.F.); biel.garcias@uab.cat (B.G.); 2Departamento Veterinaria de Laboratorio Echevarne, S.A., 08037 Barcelona, Spain; idduran@laboratorioechevarne.com; 3Catalan Wildlife Service, Centre de Fauna Salvatge de Torreferrussa, 08130 Santa Perpètua de Mogoda, Spain; rafael.molina@gencat.cat

**Keywords:** pet rabbits, antimicrobial resistance, One Health approach, zoonotic risk, Spain

## Abstract

**Simple Summary:**

Rabbits are the second most common specialty pet among households in Europe and the USA. However, research on antimicrobial resistance (AMR) in pet rabbits is very scarce. Therefore, scientific data on AMR in pet rabbits is urgently needed as a guide for veterinarian clinicians to optimize antibiotic use in rabbits for reducing the selection of antibiotic resistance. In addition, antimicrobial stewardship programs should be conducted to educate rabbit owners not to misuse antibiotics on their pets as it may put their own health at risk. This paper aims to provide an overview of the current state of AMR in rabbits attended to in veterinary clinics distributed in Spain to highlight the importance of addressing AMR under the One Health approach.

**Abstract:**

Research on antimicrobial resistance (AMR) in pet rabbits is very scarce. The aim of this study was to provide an overview of the current state of AMR in rabbits attended to in veterinary clinics distributed in Spain. Records of 3596 microbiological results of clinical cases submitted from 2010 to 2021 were analyzed. *Staphylococcus* spp. (15.8%), *Pseudomonas* spp. (12.7%), *Pasteurella* spp. (10%), *Bordetella* spp. (9.6%) and *Streptococcus* spp. (6.8%) were the most frequently diagnosed agents. Enterobacteriaceae, principally *Escherichia coli*, *Klebsiella pneumoniae* and *Enterobacter cloacae*, accounted for about 18% of the cases and showed the highest proportion of multi-drug resistance (MDR) isolates, with 48%, 57.5% and 36% of MDR, respectively. Regarding the antimicrobial susceptibility testing for a number of antimicrobial categories/families, the largest proportion of isolates showing resistance to a median of five antimicrobial categories was observed in *P. aeruginosa*, *Stenotrophomonas maltophilia* and *Burkolderia* spp. In contrast, infections caused by *Staphylococcus*, *Streptococcus* spp. and *Pasteurella multocida* were highly sensitive to conventional antimicrobials authorized for veterinary use (categories D and C). The emergence of AMR major nosocomial opportunistic pathogens such as *P. aeruginosa, S. maltophilia* and *K. pneumoniae* in pet rabbits can represent a serious public health challenge. Consequently, collaboration between veterinarians and human health professionals is crucial in the fight against antimicrobial resistance, to optimize, rationalize and prudently use antimicrobial therapies in domestic animals and humans.

## 1. Introduction

Antimicrobial resistance (AMR) is a growing global concern, with the emergence of multidrug-resistant bacteria representing a significant threat to human and animal health. The close interaction between pets and their owners can facilitate the transmission of pathogenic bacteria between humans and animals, especially multidrug-resistant (MDR) microorganisms, representing a serious threat for human and animal health. Moreover, MDR infections complicate medical management, lengthen hospital stays and have a big economic impact [1].

Rabbits are the second most common specialty/exotic pet mammals among households, according to the American Veterinarian Association, and they are considered ideal pets for children in the USA and Europe [2]. Currently, rabbits are expanding in other regions, being extremely popular pets in Australia and in Asian countries such as Japan and Singapore [3]. Pet rabbits may also host parasites (*Encephalitozoon cuniculi*, *Cryptosporidium* spp., *Giardia* spp. and *Tricostrongylus* spp.), viruses (hepatitis E), bacteria (*Bartonella* spp., *Pasteurella* spp.) and fungi (dermatophytosis), which can be potential zoonotic pathogens for humans [4]. Elder people and children younger than 5 years, as well as immunocompromised persons and pregnant women, are particularly most susceptible to such pet-induced zoonoses [5]. However, related to AMR bacteria, most of the data published in pets are focused on dogs and cats [6,7,8,9,10,11] and very few are related to other pet species such as rabbits [4,12]. 

Thus, understanding the prevalence of AMR among pet rabbits is highly necessary from both veterinary and human medicine perspectives. Since the number of antibiotics available in veterinary medicine is limited, and there are many antibiotics contraindicated for oral administration in rabbits because of their toxicity (clindamycin, lincomycin, erythromycin, ampicillin, amoxicillin/clavulanic acid and cephalosporins), it is very important to select the best therapeutic option [13,14,15]. Thus, the use of antibiotics should be based on the results of susceptibility testing and the specific needs of each rabbit case. Empiric treatment should be administered only for urgent cases where the survival of the animal is compromised and should be based on scientific evidence. Therefore, scientific data on AMR in pet rabbits is urgently needed as a guide for veterinarian clinicians to optimize antibiotic use in rabbits for reducing the selection of antibiotic resistance. In addition, antimicrobial stewardship programs will also be conducted to educate rabbit owners not to misuse antibiotics on their pets as it may put their own health at risk.

This paper aims to provide an overview of the current state of AMR in rabbits attended to in veterinary clinics distributed in Spain and discuss the potential causes and consequences of this problem under the One Health approach.

## 2. Materials and Methods

### 2.1. Database Collection and Management

Retrospective data on microbiological results of clinical cases of pet rabbits submitted between 2010 and 2021 from Spain and Portugal were analyzed. The database was comprised of 3596 records. These records were provided by a private diagnostic laboratory in Barcelona (Spain), which has had the ISO-9001 quality management system certificate since 1998, and the ENAC (National Accreditation Entity) accreditation according to criteria included in the ISO standard/IEC 17025 defined in technical annexes 511/LE1947 for pharmaceutical toxicology and microbiology tests.

The first step was to filter and categorize the study variables to homogenize all the data for performing subsequent descriptive and quantitative statistical analyzes. The following variables were included in the study: geographical location of the sample; origin of the sample classified in categories as regards the pathological relevance in rabbits (abscesses, dental disease, dermatitis/skin disease, otitis, conjunctivitis, reproductive tract, respiratory tract, urinary tract infections); microbiological result (positive identification or negative/absence of bacterial growth); bacterial species (grouped by genus and species) and the antimicrobial sensitivity results (from the 84 antibiotics included in the study, the antibiotics most conventionally used in veterinary medicine and as a last resort for human medicine were selected).

### 2.2. Microbiological Diagnosis Techniques and Antimicrobial Susceptibility Testing

Bacterial identification was performed by means of the MALDI-TOF mass spectrometer, as previously described [9,10,12]. Gram-positive bacterial isolates were found by the antimicrobial susceptibility test (AST) using the standard disk diffusion method according to Performance Standards for Antimicrobial Susceptibility Testing for bacteria isolated from animals [16] and humans [17], for monitoring resistant microorganisms as a potential risk to public health. The panel included 21 antimicrobials corresponding to 9 classes or categories, and their respective disc concentrations: β-lactams (penicillin (PEN/10U), ampicillin (AMP/10 µg), cephalexin (LEX/30 µg), cefuroxime (CXM/30 µg), cefotaxime (CTX/30 µg), cefepime (FEP/30 µg), imipenem (IMI/10 µg), amoxicillin + clavulanic acid (AMC/30 µg) and aztreonam (AZT/30 µg)), fluoroquinolones (ciprofloxacin (CIP/5 µg), enrofloxacin (ENR/5 µg), marbofloxacin (MBF/5 µg)), aminoglycosides (amikacin (AMK/30 µg) and gentamicin (GEN/10 µg)), tetracyclines (doxycycline (DOX/30 µg)), polymyxins (polymyxin B (PMB/300 µg) and colistin (COL/10 µg)), trimethoprim/sulfonamides (trimethoprim + sulfametoxazol (TxS/25 µg)), glycopeptides (vancomycin (VAN/30 µg)), phosphonates (Fosfomycin (FOS/50 µg)) and phenicol’s (chloramphenicol (CHL/10 µg)). In parallel, NM44 MicroScan (Beckman Coulter, Villepinte, France) system testing was performed to detect minimal inhibitory concentrations (MIC) [9,10]. Additionally, quality control for the AST was performed using internal controls in each automatic panel of the NM44 MicroScan (Beckman Coulter, Villepinte, France). In the case of manual antibiograms, McFarland standards were used as a reference, previously confirmed by a Densicheck (bioMérieux, Madrid, Spain).

Based on the lab testing readings, isolates were classified as susceptible, intermediate or resistant. For showing the AST histograms of antimicrobial categories, all isolates that exhibited intermediate resistance were grouped with the susceptible ones. Multidrug resistance (MDR) was defined as resistance to at least 1 agent in ≥3 antimicrobial categories and determined using R version 4.2.0 (R Core Team, 2022) [18], with the AMR package [19], as defined by Magiorakos et al. (2012), where intrinsic resistances were not considered in the analysis [20]. In the definitions proposed for MDR in this study, a bacterial isolate is considered resistant to an antimicrobial category when it is ‘non-susceptible to at least one agent in a category’ [20]. 

## 3. Results

The analysis of this study was conducted with 3596 records of clinical cases from different provinces of Spain. A microbiological identification was obtained in 2998 (83.4%) of the samples, and 598 samples were negative (no microbiological culture). According to the bacteriological identification, the most prevalent genera were *Staphylococcus* spp. (15.8%), *Pseudomonas* spp. (12.7%), *Pasteurella* spp. (10%), *Bordetella* spp. (9.6%) and *Streptococcus* spp. (6.8%). Enterobacteriaceae represented around 18% of the isolates, with *Enterobacter* spp., *Escherichia* spp. and *Klebsiella* spp. being the most frequent ones (Table 1).

The distribution of bacteria according to the origin of the samples showed that the most frequent origins were those coming from the respiratory tract (53%), followed by otitis (18%), abscesses, principally located in the head (16%), conjunctivitis (5%), reproductive tract (3%), skin disease/dermatitis (2%), urinary tract (2%) infections and dental disease (1%). 

The most frequent pathogens involved in cases of abscesses (located mainly on the head), dental disease, dermatitis/skin disease, conjunctivitis and otitis were Gram-positive cocci (principally *Staphylococcus* spp., followed by *Streptococcus* spp.) and *Pseudomonas aeruginosa* (Figure 1). *Streptococcus* spp. was the primary agent responsible for reproductive tract infections, while *Enterococcus* spp. was the most frequently responsible for urinary infections. Gram-negative infections caused by *P. multocida* and *B. bronchiseptica* (33%), followed by *P. aeruginosa* (15%), were the most frequent causes of respiratory infections. Additionally, *Pasteurella* spp. was found in cases of abscesses, dermatitis/skin disease, conjunctivitis and otitis.

The Enterobacteriaceae family, principally represented by *E. coli*, *K. pneumoniae* and *E. cloacae,* was homogeneously distributed in all the pathological categories. *Acinetobacter* spp. was also isolated from diverse origins. Other less prevalent pathogens were *Stenotrophomonas maltophilia*, isolated from respiratory and urinary infections, conjunctivitis and otitis, *Burkholderia* spp., isolated from abscesses and the urinary tract, and *Trueperella pyogenes*, found in abscesses and dental disease (Figure 1). 

As regards the AST results, *P. aeruginosa* was the most prevalent pathogen with the highest levels of AMR, presenting 80% of strains resistant to penicillins, inhibitors of β-lactamases (AMC), 1st and 2nd generation (1G/2G) cephalosporins, trimethoprim/sulfonamides and phenicols, and 60% of strains were resistant to 3rd and 4th generation (3G/4G) cephalosporins (Figure 2). As regards to the MDR profile, 8% (31/381) of *P. aeruginosa* strains were MDR, but the average number of antimicrobial categories or families that presented resistance was 5 (Table 2, Figure 3).

Other less representative bacteria but with the largest proportion of isolates showing resistance to a median of five antimicrobial categories were *Stenotrophomonas* spp., specifically *S. maltophilia* and *Burkolderia* spp. (Table 2). Both bacterial species were not considered MDR strains because of the intrinsic resistance to several families (Table 2). However, from the clinical point of view, it is interesting to remark that they presented high frequencies of resistance to β-lactams, with special attention to carbapenems (>80% *Stenotrophomonas* and 50% *Burkholderia*), also to polymyxins (>75% *Burkholderia* and 60% *Stenotrophomonas*) and fluoroquinolones (55% *Burkholderia* and 48% *Stenotrophomonas*) (Figure 2). 

The Enterobacteriaceae family, represented principally by *E. coli*, *K. pneumoniae* and *E. cloacae*, showed a high prevalence of AMR to β-lactams: penicillins (>80% *K. pneumoniae* and *E. cloacae*), AMC (>80% *E. cloacae*), 1G/2G cephalosporins (>50% *K. pneumoniae* and >70% *E. cloacae*) and 3G/4G cephalosporins (50% *K. pneumoniae*). Moreover, *K. pneumoniae* isolates showed resistance to trimethoprim/sulfonamides (50%) and to fluoroquinolones (60%) (Figure 2). Moreover, the percentage of MDR isolates was notable in enterobacteria isolates such as *K. pneumoniae* (58%), *E. coli* (48%), *Proteus* spp. (47%) and *E. cloacae* (36%) (Table 2). In addition, the average number of antimicrobial categories presenting resistance was three in almost all enterobacteria, except for *K. pneumoniae,* in which it was four (Figure 3).

Another bacterial spp. with a considerable resistance profile was *Acinetobacter* spp., with 11% of MDR (Table 2) and nearly 60% of the isolates presenting resistance to penicillins, AMC and 1G/2G cephalosporins (Figure 2). *Bordetella*, mainly *B. bronchiseptica*, was another pathogen with AMR resistance to 3 antimicrobial categories, finding 80% of resistance to penicillins and 1G/2G cephalosporins and 50% to 3G/4G cephalosporins (Figure 2). Additionally, *Enterococcus* spp., frequently isolated from UTI in rabbits, showed a high prevalence of AMR to aminoglycosides (>80%), 1G/2G cephalosporines (>70%), fluoroquinolones (>50%) and 3G/4G cephalosporines (>40%), with 6.5% of MDR strains (Figure 2, Table 2). As regards the susceptibility to vancomycin, *Streptococcus* spp. (22%) presented the highest frequency of resistance, followed by *Enterococcus* (12%) and *Staphylococcus* (6%). 

Finally, Gram-positive cocci (*Staphylococcus* and *Streptococcus*) and other Gram-negative bacteria, such as *Pasteurella multocida* and *Trueperella pyogenes*, were sensitive to a wide panel of conventional antimicrobial agents, including those classified in categories D and C (Figure 2).

Values of minimal inhibitory concentrations (MIC) can be found in Appendix A. In general, *P. aeruginosa* presented the highest levels of MIC_90_ for a major portion of the antimicrobials tested. 

## 4. Discussion

This study aimed to highlight the importance of addressing AMR in pet rabbits as a crucial step in the fight against antimicrobial resistance more broadly, enhancing the correct use of antibiotics to preserve their efficacy in the future to effectively control bacterial infections in people and pets.

The positive finding of these results is that the most common infections caused by Gram-positive cocci, basically *Staphylococcus* and *Streptococcus* spp. involved in abscesses, dental disease, dermatitis/skin disease, conjunctivitis and otitis, presented a low frequency of AMR, being sensitive to antimicrobials of categories D and C according to the EMA [21]. Additionally, *Pasteurella* (*P. multocida*), one of the most common bacteria of rabbits which colonizes the upper respiratory tract and the oro-pharynx, was found to be highly sensitive to conventional D and C class drugs. *Pasteurella multocida* can reside in the nasal flora of asymptomatic rabbits and spread to other sites during grooming, and it is also frequently isolated from abscesses because this bacterium has capsular polysaccharides that resist phagocytosis [22]. In pet rabbits, most abscesses occur around the head and face and are associated with dental disease. Another bacterial agent isolated from abscesses and dental disease was *Trueperella pyogenes*. This bacterium has been associated with sporadic cases of suppurative disorders in the lungs, liver, spleen and brain of rabbits [23]. Fortunately, and similar to *P. multocida*, *T. pyogenes* presented a highly sensitive pattern of AMR in our pet rabbits.

The zoonotic risk of *P. multocida* transmission to humans must be considered through bites, scratches or licks of companion animals, with the development of local inflammatory reactions and occasionally the occurrence of abscesses in people [5,24,25,26]. Moreover, in some patients, principally in immunocompromised people or persons with pulmonary disorders, pasteurellosis may result in more severe pathologies, such as pneumonia, endocarditis, meningitis and sepsis [27,28]. In a recent paper, *P. multocida* belonging to capsular type A was the type most often detected in humans, and although it was susceptible to the tested antibiotics, in agreement with our AST results, it was equipped with several virulence genes [4]. These findings are of particular interest because rabbits recovered from pasteurellosis very often become asymptomatic carriers of this infection and can represent a risk for the household members, especially for children and elder people [29]. 

On the other hand, Gram-negative infections caused by *P. multocida* and *B. bronchiseptica,* followed by *P. aeruginosa*, were principally involved in respiratory infections, in agreement with a pervious study conducted in pet rabbits in France [30]. In that study, the authors concluded that marbofloxacin was shown to be a potentially good treatment option for upper respiratory tract disease in pet rabbits. Although the use of fluoroquinolones is the most common therapeutic option in exotic animal medicine, the EMA recommendations appeal for the use of D and C categories in order to preserve the efficacy of critical antimicrobial classes such as fluoroquinolones (category B). For this reason, and considering the AST results of our study, for respiratory infections caused by *P. multocida* or *B. bronchiseptica*, trimethoprim/sulfonamides could be a good candidate for treatment in pet rabbits. 

Non-fermenting Gram-negative bacilli, such as *P. aeruginosa* and *Acinetobacter baumannii*, are among the major opportunistic pathogens involved in the global antibiotic resistance epidemic in human medicine [31]. Data on pet rabbits showed that the antimicrobial treatment of *P. aeruginosa* can be more complicated, since a high percentage of the isolates presented a resistance profile, including antimicrobials of category B (3G/4G cephalosporins and fluoroquinolones). This pathogen is also found in a wide spectrum of rabbit pathologies (abscesses, dental disease, dermatitis/skin disease, conjunctivitis, otitis and respiratory infections), and the treatment options are very few, limited to carbapenems and polymyxins, which are antimicrobials of category A (reserved for critical use in human medicine), but also to aminoglycosides. Since this former family is classified in category C, aminoglycosides could be the best option for treating pseudomonal infections in rabbits. It is important to note that polymyxins can be highly toxic to rabbits and should be avoided for treatments. However, this antimicrobial class was added in this study for its relevance as a last-resort drug for human medicine. 

As regards the Enterobacteriaceae family, *E. coli*, *K. pneumoniae* and *E. cloacae* represented the most frequent species isolated from a large diversity of pathologies. *Escherichia coli* infections can cause enteritis, sepsis and urinary tract infections in rabbits. Although *E. coli* was the most prevalent enterobacteria, the frequency of MDR was lower compared to *K. pneumoniae*, as observed in other pet studies in Spain [9,12]. According to our results, good candidates for treating infections caused by *E. coli* could be aminoglycosides. On the other hand, *K. pneumoniae* showed high resistance to most of the antimicrobial classes of conventional use in veterinary medicine, leaving carbapenems as the best therapeutic option even though it is a category A drug. Considering other antimicrobials authorized for veterinary medicine, the best options were aminoglycosides, chloramphenicol or doxycycline, although more than 40% of the isolates presented resistance to these drugs. As a result, the increasing occurrence of *K. pneumoniae* as a MDR infection and a zoonotic agent represents a real threat to both animal and human health [32,33]. In addition, *E. cloacae* is another emerging pathogen recognized as a nosocomial bacterium contributing to septic arthritis, skin/soft tissue infections, bacteremia, lower respiratory tract and urinary tract infection, endocarditis, osteomyelitis and intra-abdominal infections in humans [34].

Other less representative bacteria, but with a proportion of resistance to several antimicrobial categories (five as a median), were *S. maltophilia* and *Burkolderia* spp. Both bacterial species presented high frequencies of resistant isolates to β-lactams (including carbapenems), as well as to category A (polymyxins) and B (fluoroquinolones) drugs. *S. maltophilia* is an emerging nosocomial pathogen, with intrinsic resistance to beta-lactams, capable of causing healthcare-associated infections in intensive care units, life-threatening diseases in immunocompromised patients and severe pulmonary infections in cystic fibrosis and COVID-19-infected individuals [35,36]. 

Lastly, it was interesting to note that 12% of the Enterococcus isolates were resistant to vancomycin, more than 80% to aminoglycosides, around 70% to 1G/2G cephalosporines, half of them to fluoroquinolones and 40% to 3G/4G cephalosporines. With these AMR profiles, the treatment of UTI caused by this bacterium in rabbits can be difficult to plan without a previous susceptibility testing.

Overall, the emergence of AMR strains such as *P. aeruginosa*, *A. baumannii*, *S. maltophilia* and *K. pneumoniae* in pet rabbits can represent a serious health threat for the owners, since they are among the major opportunistic pathogens with significant contributions to mortality in hospitals worldwide [31,37]. Moreover, these pathogens are designated as urgent/serious threats by the Centers for Disease Control and Prevention and are part of the World Health Organization’s list of critical priority pathogens [38]. 

It is important to remember that the list of antimicrobial therapeutic options for treating bacterial infections in rabbits is not exhaustive and the use of antibiotics should be based on the results of susceptibility testing, the specific needs of each animal case and the risk of toxicity of these drugs in rabbits. However, for urgent cases, when the severity of the clinical process requires immediate antimicrobial therapy with no time for AST analysis, the data reported in the present study can be useful for veterinary practitioners to apply empirical therapy. It is crucial to keep in mind that the best way to proceed for reducing AMR selection is to perform a proper antimicrobial diagnosis with the corresponding AST. Then, antimicrobials with a sensitive result must be prioritized according to the EMA categories, mainly D and C.

Finally, the results of this study provided objective data on the microbiological results in pet rabbits in Spain. The high levels of AMR to critically important antibiotics in human medicine found in pet rabbits are of great concern since potential transmission of resistance genes from rabbits to humans or other pets can occur. Considering that the predominant bacteria in this study are among the top pathogens directly attributed to human deaths due to AMR, it is critical that veterinarians and physicians work together to optimize, rationalize and prudently use antimicrobial therapies in domestic animals and humans under the One Health approach.

## Figures and Tables

**Figure 1 vetsci-10-00352-f001:**
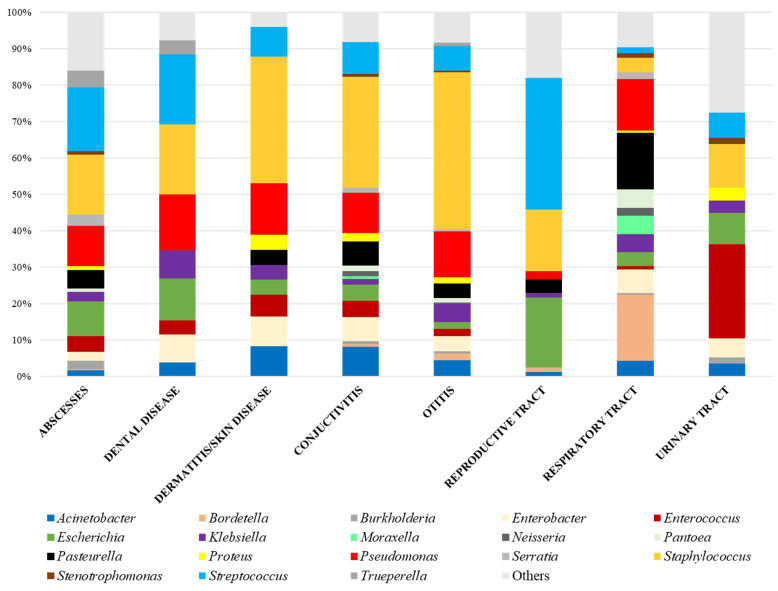
Distribution of bacteria genera regarding the sample origin in pet rabbits.

**Figure 2 vetsci-10-00352-f002:**
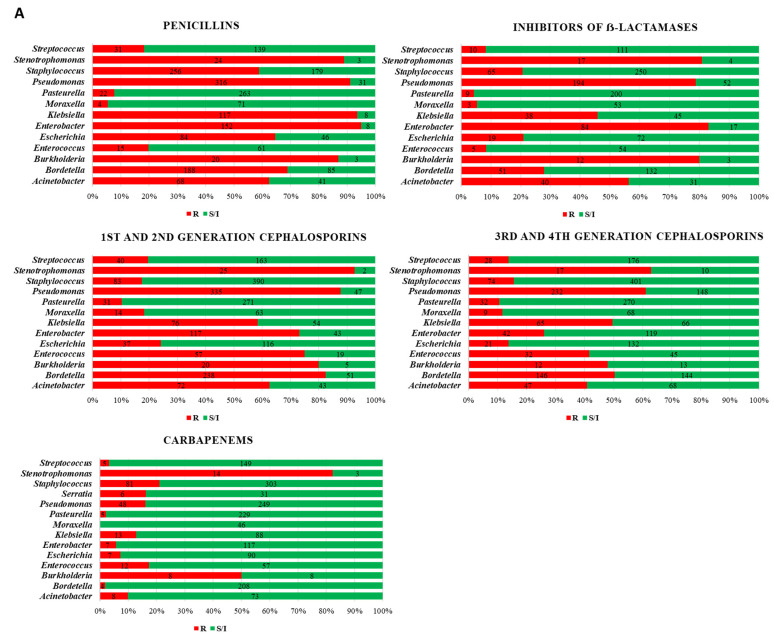
Percentage and number of isolates presenting resistant (red) or susceptible/intermediate (green) results in the AST for (**A**) the beta-lactams class and (**B**) other antimicrobial families.

**Figure 3 vetsci-10-00352-f003:**
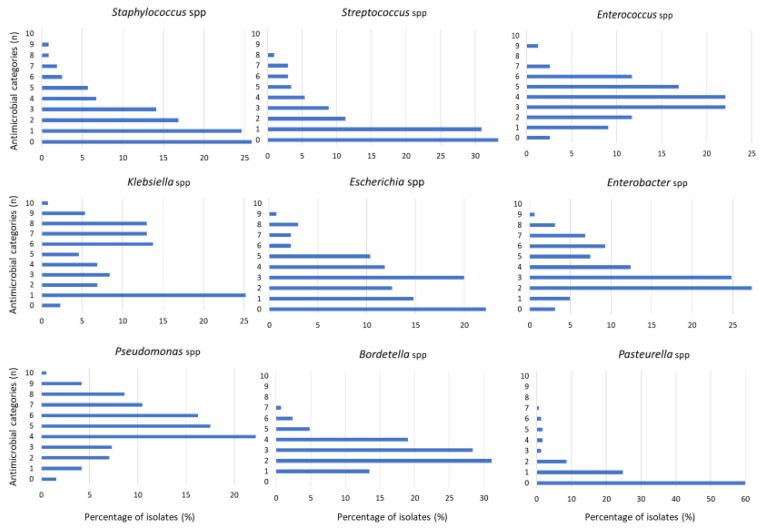
Distribution of the different bacterial isolates (%) according to the number of antimicrobial categories showing resistance.

**Table 1 vetsci-10-00352-t001:** Frequencies of bacterial species identified in pet rabbits.

Bacteria Isolates	Number (% in spp.)	Overall % (N = 2998)
***Staphylococcus* spp.**	**n = 475**	**15.8**
*S. aureus*	171 (36)	5.70
*S. xylosus*	33 (7)	1.10
*S. epidermidis*	26 (5.5)	0.86
*S. lugdunensis*	14 (3)	0.46
*S. pseudointermedius*	13 (2.8)	0.43
*S. sciuri*	9 (1.9)	0.30
*S. capitis*	9 (1.9)	0.30
*S. chromogenes*	7 (1.5)	0.23
*S. intermedius*	6 (1.3)	0.20
*S. simulans*	6 (1.3)	0.20
*S. chleiferi*	6 (1.3)	0.20
*S. cohnii*	5 (1.1)	0.16
*S. saprofhyticus*	5 (1.1)	0.16
*S. succinus*	5 (1.1)	0.16
*Others*	160 (33.7)	5.33
***Pseudomonas* spp.**	**n = 382**	**12.7**
*P. aeruginosa*	264 (69)	8.80
*P. putida*	19 (5)	0.63
*P. fluorescens*	14 (3.7)	0.46
*P. korensis*	7 (1.8)	0.23
*P. fulva*	5 (1.3)	0.16
*P. libaniensis*	5 (1.3)	0.16
*P. monteilii*	5 (1.3)	0.16
*Others*	63 (16.5)	2.10
***Pasteurella* spp.**	**n = 302**	**10.1**
*P. multocida*	230 (76.2)	7.7
*P. canis*	10 (3.3)	0.3
*Others*	62 (20.5)	2.1
***Bordetella* spp.**	**n = 289**	**9.6**
*B. bronchiseptica*	278 (96.2)	9.3
*Others*	11 (3.8)	0.4
***Streptococcus* spp.**	**n = 204**	**6.8**
*S. intermedius*	32 (15.7)	1.1
*S. anginosus*	6 (3)	0.2
*S. oralis*	6 (3)	0.2
*Others*	160 (78.4)	5.3
***Enterobacter* spp.**	**n = 161**	**5.4**
*E. cloacae*	123 (76.39)	4.10
*E. kobei*	13 (8.07)	0.43
*E. bugandensis*	6 (3.72)	0.20
*Others*	19 (11.80)	0.63
***Escherichia* spp.**	**n = 153**	**5.1**
*E. coli*	141(92.15)	4.70
*E. vulneris*	7 (4.15)	0.23
*Others*	5 (3.26)	0.16
***Klebsiella* spp.**	**n = 131**	**4.4**
*K. pneumoniae*	75 (57.3)	2.5
*K. oxytoca*	43 (32.8)	1.4
*Others*	13 (9.9)	0.4
***Acinetobacter* spp.**	**n = 115**	**3.8**
*A. iwoffii*	28 (24.4)	0.9
*A. baumannii*	16 (13.9)	0.5
*A. johnsonii*	10 (8.7)	0.3
*A. pitti*	10 (8.7)	0.3
*Others*	51(44.3)	1.7
***Pantoea* spp.**	**n = 90**	**3.0**
*P. agglomerans*	69 (76.7)	2.3
*Others*	21 (23.3)	0.7
***Enterococcus* spp.**	**n = 77**	**2.6**
*E. faecalis*	40 (51.9)	1.3
*Others*	37 (48.05)	1.23
***Moraxella* spp.**	**n = 77**	**2.6**
*M. branhamella*	32 (41.6)	1.1
*M. catarrhalis*	25 (32.5)	0.8
*Others*	20 (26)	0.7
***Serratia* spp.**	**n = 54**	**1.8**
*S. marcescens*	39 (72.2)	1.3
*S. liquefaciens*	5 (9.3)	0.2
*S. odorífera*	5 (9.3)	0.2
*Others*	5 (9.3)	0.2
***Neisseria* spp.**	**n = 37**	**1.6**
*N. animaloris/zoodegmatis*	1 (2.7)	0.03
*N. gonorrheae*	1 (2.7)	0.03
*N. species*	1 (2.7)	0.03
*Others*	34 (92)	1.1
***Proteus* spp.**	**n = 34**	**1.1**
*P. mirabilis*	28 (82.4)	0.9
*P. vulgaris*	4 (11.7)	0.1
*P. penneri*	2 (5.9)	0.1
***Trueperella* spp.**	**n = 32**	**1.1**
*T. pyogenes*	32 (100)	1.1
***Strenotrophomonas* spp.**	**n = 27**	**0.9**
*S. maltophilia*	26 (96.3)	0.9
*Others*	1 (3.7)	0.0
***Burkholderia* spp.**	**n = 25**	**0.8**
*B. cepacia*	19 (76)	0.6
*Others*	6 (24)	0.2

**Table 2 vetsci-10-00352-t002:** Average number of AMR categories and frequencies of MDR bacterial species.

Genus	Isolates	Number of AMR Categories/Families		MDR * Profile
**Gram-Negative**	**n**	**Mean**	**Median**	**%**
*Pseudomonas*	381	5.0	5	8.1
*Stenotrophomonas*	26	5.2	5	0
*Burkholderia*	25	4.6	5	0
*Acinetobacter*	115	3.0	3	11.3
*Bordetella*	289	2.8	3	0
*Pasteurella*	299	0.7	0	0
*Moraxella*	77	1.2	1	0
*Escherichia*	129	2.7	3	47.9
*Klebsiella*	134	4.4	4	57.5
*Enterobacter*	157	3.6	3	35.7
*Proteus*	34	3.1	3	47.1
*Serratia*	54	3.3	3	33.3
**Gram-Positive**	**n**	**Mean**	**Median**	**%**
*Staphylococcus*	466	2.0	1	5
*Streptococcus*	204	1.6	1	0
*Enterococcus*	77	3.7	4	6.5

* According to [20].

## Data Availability

Data will be made available upon reasonable request to the corresponding author.

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
