# Peer review of "Current Situation of Bacterial Infections and Antimicrobial Resistance Profiles in Pet Rabbits in Spain"

_vetsci, 2023, doi:10.3390/vetsci10050352_

Round 1
Reviewer 1 Report
The authors report the different bacterial species isolated from different rabbit sample evaluating their AMR profiles. The topic is interesting but, in my opinion, there is a problem related to the methods and consequently to the results. Indeed, authors doesn’t consider intrinsic resistance (expected resistant phenotype) as described by EUCAST (https://www.eucast.org/expert_rules_and_expected_phenotypes) and so the AMR profile highlighted for example for P. aeruginosa or S. maltophilia cannot be considered associated to rabbit because they are expected resistant phenotype. I suggest author to reconsider the results excluding intrinsic resistance and to reviewed the manuscript in this way.
Material and Method
Lines 92-115: I don’t understand the breakpoint used; the evaluation of colistin should be done by MIC
Results
Lines 123- 124: “A microbiological identification….598 samples had no microbiological results” what does it mean? That 598 were negative or without identification?
Fig 1: I suggest deleting the title over the figure
Lines 158-163: I suggest considering expected resistant phenotype
Line 170: please put E. cloacae and E. coli in italics
Discussion
In my opinion the discussion would be reviewed considering the results without intrinsic resistances.
Author Response
The authors report the different bacterial species isolated from different rabbit sample evaluating their AMR profiles. The topic is interesting but, in my opinion, there is a problem related to the methods and consequently to the results. Indeed, authors doesn’t consider intrinsic resistance (expected resistant phenotype) as described by EUCAST (https://www.eucast.org/expert_rules_and_expected_phenotypes) and so the AMR profile highlighted for example for P. aeruginosa or S. maltophilia cannot be considered associated to rabbit because they are expected resistant phenotype. I suggest author to reconsider the results excluding intrinsic resistance and to reviewed the manuscript in this way.
Answer: We totally agree with the reviewer observation, and we have not considered intrinsic resistance in the MDR profile analysis. Thus, we have used the following methodology:
“Multidrug resistance (MDR) was defined as resistance to at least one agent in ≥3 antimicrobial categories and determined using R version 4.2.0 (R Core Team, 2022) [15], with the AMR pack-age [16], as defined by Magiorakos, et al. (2012), where intrinsic resistances were considered in the analysis [17]. In the definitions proposed for MDR in this study, a bacterial isolate is considered resistant to an antimicrobial category when it is ‘non-susceptible to at least one agent in a category' [17].
- R Core Team (2022). R: A language and environment for statistical computing. R Foundation for Statistical Computing, Vienna, Austria. URL https://www.R-project.org/.
- Berends, M. S., Luz, C. F., Friedrich, A. W., Sinha, B. N. M., Albers, C. J., & Glasner, C. (2022). AMR: An R Package for Working with Antimicrobial Resistance Data. Journal of Statistical Software, 104(3), 1–31. https://doi.org/10.18637/jss.v104.i03
- Magiorakos, A.‐P.; Srinivasan, A.; Carey, R.B.; Carmeli, Y.; Falagas, M.E.; Giske, C.G.; Harbarth, S.; Hindler, J.F.; Kahlmeter, G.;Olsson‐Liljequist, B.; et al. Multidrug‐Resistant, Extensively Drug‐Resistant and Pandrug‐Resistant Bacteria: An International Expert Proposal for Interim Standard Definitions for Acquired Resistance. Clin. Microbiol. Infect. 2012, 18, 268–281.https://doi.org/10.1111/j.1469‐0691.2011.03570.x.
Thus, the manuscript has been reviewed taking into account the new results and a new Table 2 has been added to provide this information. We have kept the concept of number of antimicrobial categories presenting resistance and the MDR concept according to Magiorakos.
Material and Method
Lines 92-115: I don’t understand the breakpoint used; the evaluation of colistin should be done by MIC
Answer: The standard disk diffusion method was interpreted according to Performance Standards for Antimicrobial Susceptibility Testing for bacteria isolated from animals (M31‐A3, CLSI VET01, 2008) and humans (M100‐S24, CLSI, 2016) for monitoring resistant microorganisms as a potential risk to public health. This methodology has been used in our previous studies (Li et al., Frontiers in Microbiol 2021; Darwich et al., Vet Rec 2021; Muñoz et la., Animals 2022).
MIC values were also conducted for some antimicrobials and for colistin and the results of the MIC50 and MIC90 has been added as a Supplementary Table 1 in the revised version.
Moreover, we have clarified the text of this section of the M&M to avoid discrepancies.
Results
Lines 123- 124: “A microbiological identification….598 samples had no microbiological results” what does it mean? That 598 were negative or without identification?
Answer: That means they were negative with no microbiological culture. This has been clarified in the text.
Fig 1: I suggest deleting the title over the figure.
Answer: the title has been removed.
Lines 158-163: I suggest considering expected resistant phenotype
Line 170: please put E. cloacae and E. coli in italics
Answer: corrected.
Discussion
In my opinion the discussion would be reviewed considering the results without intrinsic resistances.
Answer: the discussion has been modified accordingly.
Reviewer 2 Report
This is a well written manuscript concenrning the antimicrobial resistance profiles for a wide range of bacterial pathogens that have been isolated from pet rabbit clinical cases in Spain. This issue is very interesting given the fact pet rabbits have become very popular in different countries and they live in close proximity with humans/pet owners. Moreover, the literature concerning pet rabbits is growing but still limited. This manuscript adds to the current knowledge and provides important information on the bacterial species isolated from pet rabbit clinical cases and the respective antimicrobial profiles, being a useful guide for the general practinioner that deals with pet rabbits.
The manuscript is clearly and well written, the results are presented thoroughly and in detail and the flow is nice.
Only some minor comments from my side.
Line 26: Write the scientific names in full.
Line 44: "cost money" should be written differently.
Lines 50-51: italics for the scientific names
Line 56: "(5-10)" should be in brackets if the authors refer to the respective references
Line 57: "convenient" does not seem to be relevant here. Please re-write
Line 84: maybe geographical location?
Line 90: "conventionally" maybe should be changed to "frequently"?
Line 110: "veterinary use" instead of "uses"
Line 140: maybe "was most frequently responsible"
Figure 1: Please check the title for "sample origin" instead of origen
Line 154: the authors should explain somewhere in the manuscript that 1G/2G etc refer to first and second generation for the readers that are not familiar with the term
Line 156, 162 and elsewhere in the manuscript: Please check for italics in scientific names wherever needed
Lines 199, 245 and elsewhere: When starting a sentence with a scientific name, you should use the scientific name in full e.g. Pasteurella multocida here
Line 207: Please rephrase, the beginning of the sentence does not make sense
Line 217-218: reference?
Lines 230: P. aeruginosa instead of full scientific name. Please check for the same mistake throuhghout the manuscript (e.g. Line 262).
The English language is fine. Some minor changes can be made to further improve the quality of the manuscript.
Author Response
Only some minor comments from my side.
Line 26: Write the scientific names in full.
Answer: Corrected
Line 44: "cost money" should be written differently.
Answer: we have modified the sentence as follows: MDR infections complicate medical management, lengthen hospital stays, and have a big economic impact
Lines 50-51: italics for the scientific names
Answer: We have corrected the scientific names and they are shown in italics.
Line 56: "(5-10)" should be in brackets if the authors refer to the respective references.
Answer: we have introduced the brackets.
Line 57: "convenient" does not seem to be relevant here. Please re-write
Answer: It has been modified by necessary
Line 84: maybe geographical location?
Answer: Yes, replaced
Line 90: "conventionally" maybe should be changed to "frequently"?
Answer: we prefer to keep conventionally as antibiotics are legally regulated
Line 110: "veterinary use" instead of "uses"
Answer: done
Line 140: maybe "was most frequently responsible"
Answer: done
Figure 1: Please check the title for "sample origin" instead of origen
Answer: following a reviewer consideration, the title was removed from fig 1.
Line 154: the authors should explain somewhere in the manuscript that 1G/2G etc refer to first and second generation for the readers that are not familiar with the term
Answer: Done in lines 156 and 157
Line 156, 162 and elsewhere in the manuscript: Please check for italics in scientific names wherever needed
Answer: corrected.
Lines 199, 245 and elsewhere: When starting a sentence with a scientific name, you should use the scientific name in full e.g. Pasteurella multocida here
Answer: corrected in 201 and 247 respectiviely.
Line 207: Please rephrase, the beginning of the sentence does not make sense
Answer: the sentence has been changed by this one: The zoonotic risk of P. multocida transmission to humans must be considered through bites, scratches
Line 217-218: reference?
Answer: we have included a new reference: Körmöndi S, Terhes G, Pál Z, Varga E, Harmati M, Buzás K, Urbán E. Human Pasteurellosis Health Risk for Elderly Persons Living with Companion Animals. Emerg Infect Dis. 2019 Feb;25(2):229-235. doi: 10.3201/eid2502.180641. PMID: 30666933; PMCID: PMC6346445.
Lines 230: P. aeruginosa instead of full scientific name. Please check for the same mistake throuhghout the manuscript (e.g. Line 262).
Answer: doble check it.
Reviewer 3 Report
In this manuscript by Fernandez et al., the authors present their findings on prevalence of bacterial pathogens and their antimicrobial susceptibility profiles in pet rabbits.
Major comments
1. Figure 2 can be improved. The authors must state (may be in methods) how resistance to a group of antibiotics was defined. How were the isolates defined to as resistant or sensitive, if more than one compound was tested in each group of antibiotics? for example, if an isolate was sensitive to amikacin but resistant to gentamycin, was it aminoglycosides resistant or sensitive? Also, not all antibiotics are suggested to be tested against all the isolates of gram positive and negative bacteria. For example, aztreonam has no activity against gram positive bacteria, so it does not make sense to test it against gram positive isolates. Another example is polymyxin which is used only for gram negative bacteria. It is not clear why polymyxins were tested against gram positive.
The authors also mention vancomycin was tested for gram positives. did they detect any Vancomycin resistant S. aureus or Enterococci?
2. Figure 3 needs more work on the presentation. It is not clear to the reader what the numbers on the both axes represent. The plots can also improved for better visibility, the numbers appear small and faded. Figure legend can include more description to help readers follow the figure. Figures are meant to be standalone and a reader should not have to jump back an forth from text to figure.
3. Reference is lacking for some information provided in 3rd paragraph of introduction and this appears to be the case in other places in the manuscript too. The authors definitely improve upon citing relevant references elsewhere in the manuscript too.
Minor comments
1. the authors claims rabbits as second most common pet in the US. The author should verify this information as dogs and cats seem more popular in the US.
2. some color choices in figure are too similar and does not help reader follow (example, for both Moraxella and Stenotrophomonas, it appears brown).
3. In some places the bacterial genus and species names are not italicized.
4. Figure 2. some antibiotics are spelled wrong.
The authors should consider editing for language.
Author Response
Major comments
- Figure 2 can be improved. The authors must state (may be in methods) how resistance to a group of antibiotics was defined. How were the isolates defined to as resistant or sensitive, if more than one compound was tested in each group of antibiotics? for example, if an isolate was sensitive to amikacin but resistant to gentamycin, was it aminoglycosides resistant or sensitive?
Answer: We have detailed these issues in the M&M as follows: we have used the following methodology:
“Multidrug resistance (MDR) was defined as resistance to at least one agent in ≥3 antimicrobial categories and determined using R version 4.2.0 (R Core Team, 2022) [15], with the AMR pack-age [16], as defined by Magiorakos, et al. (2012), where intrinsic resistances were considered in the analysis [17]. In the definitions proposed for MDR in this study, a bacterial isolate is considered resistant to an antimicrobial category when it is ‘non-susceptible to at least one agent in a category' [17].
Also, not all antibiotics are suggested to be tested against all the isolates of gram positive and negative bacteria. For example, aztreonam has no activity against gram positive bacteria, so it does not make sense to test it against gram positive isolates. Another example is polymyxin which is used only for gram negative bacteria. It is not clear why polymyxins were tested against gram positive.
Answer: we totally agree with the reviewer reflection. The graph of polymyxins has been modified in Fig 2 (now gram positives are excluded)
The authors also mention vancomycin was tested for gram positives. did they detect any Vancomycin resistant S. aureus or Enterococci?
Answer: These results have been included in the text: 193-195: As regards to the susceptibility against to vancomycin, Streptococcus spp (22%) presented the highest frequency of resistance, followed by Enterococcus (12%) and Staphylococcus (6%).
- Figure 3 needs more work on the presentation. It is not clear to the reader what the numbers on the both axes represent. The plots can also improved for better visibility, the numbers appear small and faded. Figure legend can include more description to help readers follow the figure. Figures are meant to be standalone and a reader should not have to jump back an forth from text to figure.
Answer: thank you for the comment. We have improved the presentation of fig 3 adding the missing information.
- Reference is lacking for some information provided in 3rd paragraph of introduction and this appears to be the case in other places in the manuscript too. The authors definitely improve upon citing relevant references elsewhere in the manuscript too.
Answer: we have included new references throughout the manuscript.
Minor comments
- the authors claims rabbits as second most common pet in the US. The author should verify this information as dogs and cats seem more popular in the US.
Answer: We have added new references and detailed that it was in the category of exotic pets:
Rabbits are the second most common specialty/exotic pet mammals among households, according to the American Veterinarian Association, and they are considered ideal pets for children in USA and Europe [2].
[2] https://www.avma.org/resources-tools/reports-statistics/us-pet-ownership-statistics (accessed on 26 april 2023).
- some color choices in figure are too similar and does not help reader follow (example, for both Moraxella and Stenotrophomonas, it appears brown).
Answer: colors have been changed
- In some places the bacterial genus and species names are not italicized.
Answer: doble check it throughout the manuscript
- Figure 2. some antibiotics are spelled wrong.
Answer: corrected.
Reviewer 4 Report
Fernández et al. performed a study to provide an overview of the current status of antimicrobial resistance in rabbits followed by veterinary clinics distributed in Spain. The results suggested that the high levels of AMR to critically important antibiotics for human medicine found in pet rabbits are of great concern, as potential transmission of resistance genes from rabbits to humans or other pets may occur. This study aims to highlight the importance of addressing AMR in pet rabbits as a crucial step in the fight against AMR more generally by improving the proper use of antibiotics to preserve their efficacy in the future to control bacterial infections in people and pets effectively as part of the One Health approach.
The manuscript was well organized and presented, and it has a clear microbiological description of the current status of antimicrobial resistance in rabbits followed by veterinary clinics distributed in Spain.
The article is well written, I have no comments, only one minor comment
· Your work demonstrates the value of monitoring the current status of antimicrobial resistance in rabbits to detect increasing resistance. It would be appropriate to include in the discussion of the work your thoughts on possible approaches that could be implemented to reduce this risk.
· Check that references have consistent formatting as indicated by the journal .
Author Response
- Your work demonstrates the value of monitoring the current status of antimicrobial resistance in rabbits to detect increasing resistance. It would be appropriate to include in the discussion of the work your thoughts on possible approaches that could be implemented to reduce this risk.
Answer: he have added a new paragraph at the end of the discussion: Lines 307-310:
It is crucial to have in mind that the best way to proceed for reducing AMR selection is to perform a proper antimicrobial diagnosis with the corresponding AST. Then, antimicrobials with a sensitive result have to be prioritized according to the EMA categories, mainly D and C.
- Check that references have consistent formatting as indicated by the journal .
Answer: References have checked and edited following the journal format.
Round 2
Reviewer 3 Report
1. Was intrinsic resistance included in the analysis? The authors mention in the line 122 that intrinsic resistance was considered. If that is the case, this is not line with the reference that the authors cite (Magiorakos et al., 2012). The cited papers states "When a species has intrinsic resistance to an antimicrobial category, that category must be removed from the list in this table prior to applying the criteria for the definitions and should not be counted when calculating the number of categories to which the bacterial isolate is non-susceptible."
At the moment, I am confused by authors explanation about inclusion of intrinsic resistance. If intrinsic resistance was included, it not appropriate to define AMR based on intrinsic resistance. The authors need to clarify this.
2. Still no references in the 3rd paragraph of introduction (line 60-72). There are some information in this paragraph that require citation.
The authors should consider English editing
Author Response
- Was intrinsic resistance included in the analysis? The authors mention in the line 122 that intrinsic resistance was considered. If that is the case, this is not line with the reference that the authors cite (Magiorakos et al., 2012). The cited papers states "When a species has intrinsic resistance to an antimicrobial category, that category must be removed from the list in this table prior to applying the criteria for the definitions and should not be counted when calculating the number of categories to which the bacterial isolate is non-susceptible."
At the moment, I am confused by authors explanation about inclusion of intrinsic resistance. If intrinsic resistance was included, it not appropriate to define AMR based on intrinsic resistance. The authors need to clarify this.
Answer: we apology the confusion. We considered the intrinsic resistance to remove from the analysis. Thus, to avoid misunderstanding we have changed the sentence as follows:
“Multidrug resistance (MDR) was defined as resistance to at least one agent in ≥3 antimi-crobial categories and determined using R version 4.2.0 (R Core Team, 2022) [15], with the AMR package [16], as defined by Magiorakos, et al. (2012), where intrinsic resistances were not considered in the MDR analysis [17].
- Still no references in the 3rd paragraph of introduction (line 60-72). There are some information in this paragraph that require citation.
Answer: new references [13-15] have been added and the numbering of citation modified accordingly:
- Carpenter JW (ed.). Exotic Animal Formulary. 3rd ed. St. Louis, Elsevier Publishing, 2005. 564 pp.
- Ivey ES, JK Morrisey. Therapeutics for rabbits. Vet Clin N Am, 2000; 3(1): 183 220.
- Quesenberry KE, JW Carpenter (eds). Ferrets Rabbits, and Rodents: Clinical Medicine and Surgery. Philadelphia, WB Saunders Co, 2004. 461 pp